# Early Sensory Profile in Autism Spectrum Disorders Predicts Emotional and Behavioral Issues

**DOI:** 10.3390/jpm12101593

**Published:** 2022-09-27

**Authors:** Maddalena Fabbri-Destro, Federica Maugeri, Carolina Ianni, Sofia Corsini, Erica Di Stefano, Stefano Scatigna, Giulia Crifaci, Gianina Bruzzi, Stefano Berloffa, Pamela Fantozzi, Adriana Prato, Rosy Muccio, Elena Valente, Susanna Pelagatti, Edoardo Pecchini, Filippo Zulli, Renata Rizzo, Annarita Milone, Valentina Viglione, Rita Barone, Gabriele Masi, Antonio Narzisi

**Affiliations:** 1Institute of Neuroscience, National Research Council of Italy (CNR), 43125 Parma, Italy; 2Child Neurology and Psychiatry Section, Department of Clinical and Experimental Medicine, University of Catania, 95124 Catania, Italy; 3Sanstefar Outpatient Rehabilitation Center, 64011 Alba Adriatica, Italy; 4Department of Child Psychiatry and Psychopharmacology, IRCCS Stella Maris Foundation, 56128 Pisa, Italy; 5Department of Computer Science, University of Pisa, Largo Pontecorvo 3, 56127 Pisa, Italy; 6Research Unit of Rare Diseases and Neurodevelopmental Disorders, Oasi Research Institute, IRCCS, 94018 Troina, Italy

**Keywords:** autism spectrum disorder, preschoolers, sensory processing measure (SPM), Child Behavior Checklist 1½-5 (CBCL 1½-5), language disorder

## Abstract

Background: Abnormal sensory reactivity is considered one of the diagnostic criteria for autism spectrum disorders (ASD) and has been associated with autism severity, poorer functional outcomes, and behavioral difficulties across the lifespan. Its early characterization could provide valuable insights into the processes favoring the instantiation of maladaptive behaviors. Objectives: The present study has two aims: (1) to describe the sensory profile of preschool children with ASD compared with an age-matched population of children with a diagnosis of language disorder (DLD) and typically developing (TD) control peers; (2) to explore within each group whether the sensory alterations play a predictive role in the instantiation of emotional and behavioral issues. Methods: The parents of 42 ASD, 18 DLD, and 56 TD filled out the Sensory Processing Measure—Preschool (SPM-P). To gather information on competencies, behaviors, and emotional problems of children, the Child Behavior Checklist 1½-5 (CBCL 1½-5) was also administered. Results: On the SPM-P, ASD and DLD samples generally had scores more compromised than control peers. The contrast between ASD and DLD was reflected in a higher (and highly significant) impairment on the social participation and hearing subscales, suggesting a greater sensitivity and a possible specificity of these scores for ASD. More importantly, linear regression analyses revealed a strong and predictive association for ASD children with SPM total scores explaining more than 50% of the variance of the CBCL 1½-5 total scores (*p* < 0.001). Conclusions: Our findings reinforce the need to detect the abnormal sensory profiles of ASD already at an early stage and during clinical evaluations. Due to the impact on the emotional and behavioral manifestations, such a procedure has significant clinical and social implications, potentially guiding the development of new interventions relying on multisensory strategies.

## 1. Introduction

Sensory processing (SP) is the modality in which sensory information (i.e., visual, auditory, vestibular, or proprioceptive stimuli) is managed by the cerebral cortex and brainstem to promote adequate responses to the environment and engagement in daily life activities. [1]. SP theory suggests that optimal functioning in everyday life requires efficient reception and integration of incoming sensory stimuli [2,3]. In turn, defective sensory processing might lead to impairments in accomplishing daily activities [4].

Atypical responses to sensory stimuli are consistently reported in autism spectrum disorders (ASD). Many studies showed elevated rates of sensory symptoms in ASD preschoolers [5] and school-aged children [6,7,8,9,10,11,12] compared with chronologically age-matched typically developing ones, and these symptoms can impact children’s participation in daily activities [13] and play skills [14]. Abnormalities occur across all sensory domains, including tactile, vestibular, auditory, and visual [15,16], and in the absence of peripheral dysfunctions such as visual or hearing loss [17].

Whether sensory abnormalities can be considered unique to ASD or present in other developmental disorders is a matter of debate [10,18]. The specificity of sensory abnormalities can be tackled along two independent axes. On one side, one can investigate the overall severity of sensory abnormalities. For instance, children with ADHD [19], fragile X, and other developmental delays [20] were shown to have higher sensory symptom rates than typical peers. Using parent-report questionnaires, Lord [21] and Rogers et al. [20] found that preschoolers with ASD have greater sensory symptoms than children with non-specific developmental delays, in contrast with a previous study by Stone and Hogan [22]. Thus, whether a peculiarity of children with ASD (also compared with other developmental disorders) is an exceedingly high rate of sensory abnormalities remains to be assessed. 

A second aspect that deserves attention is the relationship between SP and the development of secondary abnormalities with SP severity, which, even if homogeneous across different populations, could show a different capacity to predict the severity of abnormal behavior. Such a finding would position sensory abnormalities nearer to the core of the syndrome-related symptoms. From this perspective, it is intriguing to report previous theories hypothesizing that the dysfunction in processing sensory information characteristic of ASD results in adopting aberrant behaviors to make sense of and regulate stimulation from the environment [23,24,25]. Furthermore, previous studies highlighted a possible relationship between abnormal SP and anxiety disorders [26,27,28].

An advantage of such an approach is that sensory deficits can be investigated already since preschool age, thanks to the administration of parent questionnaires (e.g., Sensory Processing Assessment [29]; Sensory Profile-2 [30]; Sensory Processing Measure [31]). Beyond assisting the diagnostic pathway, early characterization of sensory abnormalities would be of tremendous value for guiding therapeutical interventions, tuning the daily sensory inputs according to the individual profile, and minimizing behavioral irritability since early life. 

Starting from these premises, we aimed to describe the sensory profile of preschool children with ASD compared with an age-matched population of children with a Diagnosis of Language Disorder (DLD) and typically developing (TD) control peers. In addition, we explored within each group whether the sensory alterations play a predictive role in the degree of emotional and behavioral abnormalities as indexed via a standardized behavioral checklist. 

## 2. Materials and Methods

### 2.1. Participants

One hundred sixteen Italian children were involved in the study. The sample was composed of 42 children (11f, 31m) who had received a diagnosis of autism spectrum disorders (ASD) (mean age = 4.0 years; SD = 1.1), 18 (2f, 16m) with a diagnosis of language disorder (DLD) (mean age = 4.4 years; SD = 0.82), and 56 typically developing children (TD) (mean age = 3.7 years; SD = 1.0). Participants with ASD and DLD were recruited at the Department of Child Psychiatry and Psychopharmacology of Istituto di Ricovero e Cura a Carattere Scientifico (IRCCS) Stella Maris (Pisa), the Department of Clinical and Experimental Medicine of the University of Catania, and the Sanstefar Abruzzo Pescara—Outpatient Rehabilitation Center. TD children were recruited in kindergartens in Perugia as children matching the clinical groups regarding age. Only the parents of TD children received a brief questionnaire about their children’s overall health condition and clinical history. Children with a history of neurological or psychiatric disorders were not enrolled, and those whose teachers’ expressed concerns about their development. 

For the ASD population, inclusion criteria were: (a) diagnosis of autism spectrum disorder according to DSM-5 [32]; (b) age 2–5 years. Exclusion criteria consisted of (a) the presence of any other axis I mental disorder or (b) a history of traumatic brain injury or any other neurological disorder as their medical record. 

According to DSM-5, language disorder (see DSM-5 315.39 (F80.9)) [32] is an impairment in processing linguistic information that affects an individual’s ability to receive and/or express language. The disorder involves persistent difficulties in the comprehension or production of spoken, written, sign language, or other forms of language. For the LD population, inclusion criteria were the presence of LD diagnosis and age ranging from 2–5 years old, speech articulation impairment due to: aphasia, apraxia, hearing loss, mental retardation; history of traumatic brain injury or any other neurological disorder as their medical record; comorbidity with ASD were considered as exclusion criteria. 

All subjects were native Italian speakers. Informed written consent was obtained from the parents of all participants. The study was approved by the Pediatric Ethics Committee of the Tuscany Region (Approval number: ACCESS 144/21), by the local ethics committee at University Hospital Policlinico Catania and was conducted according to ethical standards of the Declaration of Helsinki and the Italian Association of Psychology (AIP).

### 2.2. Procedures

The two clinical populations were evaluated using standard clinical tests. Particularly, the ADOS-2 [33] was exclusively adopted for the ASD group, administered by ADOS research reliable examiners. Moreover, for the DLD group, an ASD diagnosis was excluded by child psychiatrists and psychologists, supported by scores under 20 on the Childhood Autism Rating Scale (CARS) [34] and/or scores under eight on the Social Communication Questionnaire [35]. 

The parents of all participants filled out the Sensory Processing Measure—Preschool (SPM-P) [36]. The Home Form (adopted for this study) produce eight norm-referenced standard subscale scores: social participation (SOC), vision (VIS), hearing (HEA), touch (TOU), body awareness (BOD) (refers to proprioception), balance and motion (BAL) (refers to vestibular function), planning and ideas (PLA) (relates to praxis), and Total Sensory Systems (TOT). 

The SPM-P includes 75 items rated on the basis of the frequency of easily observable behaviors. Raters observe the child in the environment being rated for at least one month, but the child does not need to be present. The test produces a T-score for each SPM-P scale characterizing the child’s status in descriptive terms (Typical, Some Problems, or Definite Dysfunction). The SPM-P and SPM can be used for evidence-based practice, scientifically based research, differentiated instruction, and progress monitoring.

Beyond the SPM-P, the Child Behavior Checklist 1½-5 (CBCL 1½-5) was administered to parents to gather information on children’s competencies, behaviors, and emotional problems. The CBCL 1½-5 [37,38] is a 100-item parent-report measure that records preschoolers’ behavioral problems. Each item refers to a specific behavior, and the parent is requested to rate its frequency. The scoring gives a summary profile (including internalizing, externalizing, and total problems scores), a syndrome profile (emotionally reactive, anxious/depressed, somatic complaints, withdrawn, sleep problems, attention problems, and aggressive behavior), and five different DSM-oriented scales (affective problems, anxiety problems, pervasive developmental issues, attention deficit/hyperactive problems and oppositional defiant problems). 

### 2.3. Data Analysis

Data analysis comprised two main stages. In the first one, we applied an ANCOVA to compare the SPM total score and sub-scores across the investigated populations. To this aim, we considered the scores of SPM as the dependent variable and the diagnosis (ASD, DLD, TD) as the categorical independent variable. The subjects’ gender was also included as a categorical factor and age as a covariate. In this way, we tested the main and interacting effects of diagnosis after controlling for the possible effects of age and gender variables. In the case of significant effects, post hoc comparisons were Bonferroni corrected.

Subsequently, starting from the hypothesis that early sensory issues may worsen the severity of later behavior and emotional problems, we applied linear regression analyses to test whether the SPM total scores predict the behavioral abnormalities obtained by CBCL 1½-5 total scores. In case of significant results, we further analyze the link between SPM total score and internalizing and externalizing sub-scales, which are particularly relevant from a developmental and clinical point of view. The linear regression analysis considered CBCL 1½-5 total score the dependent variable and SPM total scores as a continuous regressor. The analysis was conducted on each group individually. 

## 3. Results

An initial one-way ANOVA was conducted to ascertain the age balance across groups. No significant effect of group emerged (*p* = 0.913). When submitted to ANCOVAs, all SPM subscales highlighted a similar pattern, i.e., a highly significant main effect of the group (all F ≥ 5, all *p* < 0.01). No significant effect involving Gender or Age was found. Thus, in the following, we will examine the different configurations of SPM subscales across the three groups, as witnessed by post hoc comparisons. Four scenarios emerged, as indicated by symbols in Figure 1. 

The most frequent one is indicated by circles and relates to subscales in which the two clinical populations detach from controls without differentiating from each other. This picture applies to the SPM total score as well as to the Vision, Body Awareness, Balance and Motion, Planning and Ideas subscales, with highly significant contrasts between TD and other groups (all *p* < 0.001, except for the case of Balance and Motion, showing milder yet significant difference at *p* < 0.02). 

A second scenario is met exclusively in the Hearing subscale (triangle in Figure 1), on which ASD children showed higher scores compared with both DLD and TD children (all *p* < 0.001), suggesting some degree of specificity of this function for autistic spectrum disorders. Even more interestingly, the Social Participation subscale (square in Figure 1) exhibited complete segregation among the three groups (all *p* < 0.02), with ASD presenting the highest scores, TD the lowest, and DLD set in between. While it is not surprising that both our clinical groups are more compromised than control peers in the social domain, the higher (and highly significant) impairment of ASD children compared to DLD children suggests a better sensitivity and a possible specificity of this subscale for autistic spectrum disorders. Finally, a less clear pattern results from the analysis of the Touch subscale in which the three groups still follow the same distribution but with lower distances and higher variability, resulting in only one significant post-hoc comparison between ASD and TD children. All SPM results are reported in Table 1.

Linear regression analyses are reported in Figure 2. In the control population, SPM total scores have no predictive power on the CBCL 1½-5 total score (R^2^ = 0.001, *p* = 0.77). Even if the lower ranges of the two scales of TD may have limited this analysis, no association emerged within the residual variability. Different is the case for the two clinical populations. ASD children showed a robust association between the two scales, with SPM scores explaining more than 50% of the variance of the CBCL 1½-5 total scores (*p* < 0.001). A milder and almost significant association emerged for DLD children, but SPM scores explained only 18% of the variance of the CBCL 1½-5 total scores (*p* = 0.07). In principle, these results suggested a link between the sensory issues and behavioral problems valid for both clinical populations. However, the different amount of explained variance (52 vs. 18), degree of significance (*p* < 0.001 vs. *p* = 0.07), and slope of the linear trend (0.89 vs. 0.39) indicate a strong and predictive association standing for ASD children, while only a marginal one for DLD children. 

We then repeated the same procedures for internalizing and externalizing subscales for the two clinical populations. SPM total scores successfully predicted these domains for both groups (see Figure 3), suggesting an even stronger relationship between sensory issues and clinically relevant emotional and behavioral scales. 

## 4. Discussion

In the present study, we investigated the sensory profiles of preschool children with ASD compared with an age-matched population of DLD and with typically developing control peers. Abnormal sensory profiles in ASD are today recognized in the DSM-5, confirmed across the lifespan [39,40] and cross-culturally [41], demonstrating their primary importance in the neurobiology of the disorders [42]. In addition, starting from the notion that abnormal sensory processing might favor the instantiation of maladaptive behavior [43], we further explored within each group whether the sensory alterations predict the degree of emotional and behavioral abnormalities as indexed via a standardized child behavior checklist 1½-5 (CBCL 1½-5). 

Before discussing the results, it is worth explaining why we selected DLD as a second sample of neurodevelopmental disorders. Language and communication deficits characterize autism spectrum disorders and developmental language disorders, so previous studies advanced the hypothesis that they may have different manifestations of the same underlying cause [44]. DLD is a heterogeneous disorder whose symptoms can be either expressive, receptive, or a combination of the two [45]. Having trouble with language means that children with DLD may have difficulty socializing with their peers, talking about how they feel, and learning in school [46]. In parallel, ASD is a genetically-based disorder characterized primarily by social deficits. One of the two main diagnostic features (impaired communication and reciprocal social interaction) lies within the social domain. Thus, testing both these populations and comparing their results should let us understand whether the two neurodevelopmental disorders (similar in the socio-communicative impairments but different in terms of etiogenesis and neurobiology) determine the similar sensory profile, detailing the subscales presenting a high degree of similarity or segregation. As a side note, the young age of our populations brings added value to our study as it limits the probability of encountering secondary symptoms and positions our participants in a time window known to be critical for the development of later cognitive and social abilities [47].

The general picture from our results indicates a similar pattern for both clinical populations at the SPM total score and in most subscales, not surprisingly, with scores more compromised than control peers. Thus, one could conclude that the two neurodevelopmental disorders overlap in sensory abnormalities and that the SPM is sensitive to but not specific to ASD. However, when examining the individual subscales, a higher (and highly significant) impairment for ASD relative to DLD was found in the Social Participation and Hearing subscales (see also [24]), suggesting a larger sensitivity and a possible specificity of these scores for autistic spectrum disorders. In summary, the extent of sensory abnormalities between ASD and DLD appears comparable. Yet, specific features for ASD children suggest that sensory impairments are not an after-effect of communication disorders but primarily linked to ASD ones. While we cannot be conclusive about this interpretation (given the explorative nature of our study), we believe that our findings offer insights into the portions of sensory deficits more tightly linked to the neurobiology of ASD.

A strong element adding to the specificity of SPM for ASD stems from the finding that the overall amount of sensory deficits in ASD children predicts behavioral abnormalities much better than for DLD. The behavioral predictability of SPM can be observed from two different perspectives: on one side, it stands only for clinical populations but not for TD, indicating that sensory abnormalities lead to behavioral abnormalities across multiple neurodevelopmental conditions [48,49]. On the other side, very distinct patterns emerge even between the two clinical populations. Indeed, the different amount of explained variance (52 vs. 18), degree of significance (*p* < 0.001 vs. *p* = 0.07), and slope of the linear trend (0.89 vs. 0.39) indicate a strong and predictive association standing solidly for ASD children, while only marginally for DLD ones. This result is even more relevant given the subtle (and not significant) difference between the two clinical populations at the SPM total scores, indicating that the stronger association is not due to a different range of the regressor but rather to a deeper link between sensory abnormalities and emotional and behavioral issues positioning the first nearer to the factors instantiating and modulating the severity of abnormal behavior.

Before drawing conclusions, it is worth noting some limitations concerning our study. On one side, the DLD population had a different size compared with the other two groups; however, the strength of the statistical findings seems to temper the relevance of this aspect. More important is the impact of testing only one clinical population beyond ASD. Being conclusive about the specificity/generalizability of our findings would require sampling multiple clinical conditions within the spectrum of neurodevelopmental disorders. 

## 5. Conclusions

Together, these findings reinforce the need to study sensory features of ASD already at an early stage, also due to their impact on the later behavioral manifestations. Identifying abnormal sensory profiles in ASD has significant clinical and social implications, potentially guiding the development of interventions for improving their social participation. The reactivity to sensory stimuli is reported in the DSM-5, demonstrating its primary importance in describing the syndrome. Atypical sensory processing in ASD is postulated to be one of the core features of ASD that can interfere with development, impede participation in everyday activities [50], and create a maladaptive developmental trajectory of functional impairment [51].

To date, interventions targeting sensory abnormalities (e.g., DIR/Floor-Time) [52] or sensory integration therapy [53] emphasize the relation between sensory experiences and motor and emotional/behavioral performance [54]. The intervention strategies encompass planned and controlled sensory experiences, emphasizing the production of functional and adaptive responses to sensory stimuli. To integrate our findings with this view, one could develop individualized treatments according to each child’s sensory profile, evaluating the adoption of sensory integration strategies assisting the administration of the deficient modalities with the spared ones. Even a small gain in sensory processing should bring benefits in terms of emotional and behavioral profile.

## Figures and Tables

**Figure 1 jpm-12-01593-f001:**
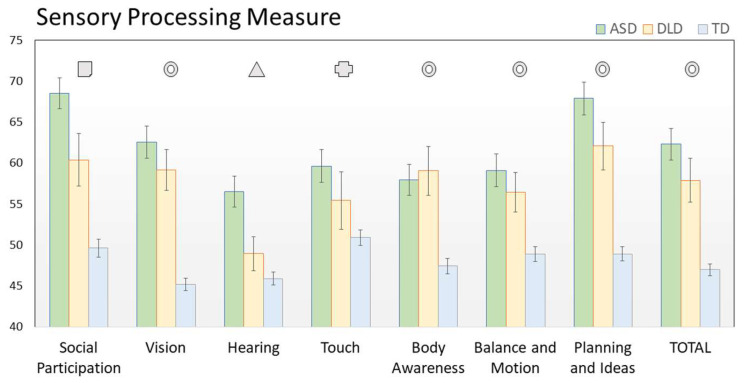
The figure reports the results of SPM subscales across the three groups (ASD in green, DLD in orange, and TD in cyan). Each symbol indicates the pattern emerging by post-hoc comparisons. Circles relate to the subscales in which ASD and DLD detach from TD without differentiating. Triangles indicate the subscales in which ASD children have higher scores than DLD and TD children. Squares indicate the subscales in which all groups differ from each other. Crosses indicate the subscales where only the post-hoc comparison between ASD and TD children was significant.

**Figure 2 jpm-12-01593-f002:**
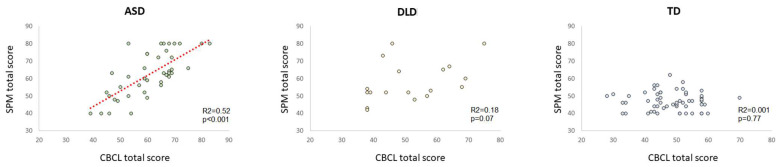
The left panel reports the linear regression analysis between SPM total scores and CBCL 1½-5 total scores for ASD children. Dots represent individual participants; the significant linear trend is reported as the red dotted line. R and *p* values are indicated on the bottom right. The middle and the right panel’s report the same data for DLD and TD groups. For the convenience of the reader, y-axes have been kept homogeneous.

**Figure 3 jpm-12-01593-f003:**
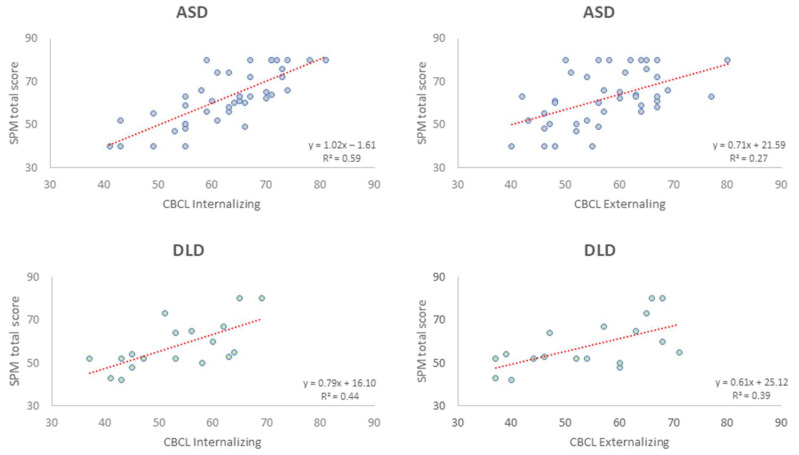
The figure reports the linear regression analysis between SPM total scores and CBCL 1½-5 Internalizing and Externalizing scores for ASD and DLD groups. Dots represent individual participants; in the case of significant regression, the linear trend is reported as the red dotted line. R and *p* values are indicated in each panel. For the reader’s convenience, x- and y-axes have been kept homogeneous.

**Table 1 jpm-12-01593-t001:** Means of the SPM subscales and total score, along with the pair-wise *p* values (*p* < 0.05, *p* < 0.001).

SPM Subscale	ASD	DLD	TD	ASD vs. TD	ASD vs. DLD	DLD vs. TD
Social Participation	68.55	60.39	49.64	** *p* ** **< 0.001**	** *p =* ** **0** **.024**	** *p <* ** **0** **.001**
Vision	62.57	59.17	45.23	** *p <* ** **0** **.001**	*p =* 0.617	** *p <* ** **0** **.001**
Hearing	56.52	48.94	45.91	** *p <* ** **0** **.001**	** *p =* ** **0** **.011**	*p =* 0.663
Touch	59.64	55.44	50.91	** *p <* ** **0** **.001**	*p =* 0.518	*p =* 0.378
Body Awareness	57.95	59.06	47.43	** *p <* ** **0** **.001**	*p =* 1.000	** *p <* ** **0** **.001**
Balance and Motion	59.12	56.44	48.93	** *p* ** **< 0** **.001**	*p =* 1.000	** *p =* ** **0** **.019**
Planning And Ideas	67.91	62.11	48.93	** *p <* ** **0** **.001**	*p =* 0.140	** *p <* ** **0** **.001**
Total	62.31	57.89	46.98	** *p <* ** **0** **.001**	*p* = 0.307	** *p* ** **< 0** **.001**

## Data Availability

The data presented in this study are available on request from the corresponding authors. The data are not publicly available due to ethical restrictions.

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
