# Peer review of "Early Sensory Profile in Autism Spectrum Disorders Predicts Emotional and Behavioral Issues"

_jpm, 2022, doi:10.3390/jpm12101593_

Round 1
Reviewer 1 Report
INTRODUCTION
The introduction is well written and covers almost all known aspects of ASD and sensory processing; it is missing mention of the relationship of sensory issues to the child's daily functioning. I suggest including some references.
Ahmed, S., Waseem, H., Sadaf, A., Ashiq, R., Basit, H., & Rose, S. (2021). Daily Living Tasks Affected by Sensory and Motor Problems in Children with Autism Aged 5-12 Years.
Yela-González, N., Santamaría-Vázquez, M., & Ortiz-Huerta, J. H. (2021). Activities of Daily Living, Playfulness and Sensory Processing in Children with Autism Spectrum Disorder: A Spanish Study. Children, 8(2), 61.
METHODS
Participants
Some comments related to Participants section:
· IRCCS ¿meaning?
· Include more data on the location of the study. Country?
· Type of recruitment? Randomized, snowball, which?
· “Parents were given a brief questionnaire about their children's overall health condition and clinical history” It is not clear whether this phrase is for all parents or only for TD children.
· Related to DLD population: This population was not diagnosed based on the DSM-5? Clarify this point whether they did or did not. If they did not, avoid talking about DLD, and talk about language problems, without mentioning the diagnosis.
· Was any analysis performed to confirm that the samples were age-matched?
Procedures section
First paragraph: All of these scales seem not to have been used in the study, so I suggest that they be eliminated. If the administration of any of them was used for some type of analysis that is not captured in the study, this analysis should be included.
Data analysis section
Typographical error. Font size change in the middle of the paragraph. “and emotional problems, we applied linear regression analyses”
RESULS
Figure 1: DLSD: error.
It would be more interesting to add a table with the numerical results of this analysis. They provide more data than graphs and help to better understand the differences.
Figure 2 Organize the wording of the text according to the organization of the images. Start the text with the first of the three graphics.
DISCUSION
The limitations of the study have not been included
CONCLUSION
The conclusion should not introduce references. It should be the final point of the work, highlighting the main findings, establishing the connection with clinical practice. I suggest that it be rewritten. Part of the text that has citations could be included in the discussion.
“one could develop individualized treatments according to each child's sensory profile, evaluating the adoption of multisensory strategies”
This statement denotes that the authors' knowledge of sensory integration therapy is shallow. Multisensory stimulation is a strategy that differs greatly from sensory integration, which is probably what the authors want to emphasize. I suggest they avoid talking about multisensory stimulation and talk about sensory integration therapy.
REFERENCES
Review this section. There are some errors. View reference n24.
Author Response
REVIEWER 1
Q1: The introduction is well written and covers almost all known aspects of ASD and sensory processing; it is missing mention of the relationship of sensory issues to the child's daily functioning. I suggest including some references.
- Ahmed, S., Waseem, H., Sadaf, A., Ashiq, R., Basit, H., & Rose, S. (2021). Daily Living Tasks Affected by Sensory and Motor Problems in Children with Autism Aged 5-12 Years.
- Yela-González, N., Santamaría-Vázquez, M., & Ortiz-Huerta, J. H. (2021). Activities of Daily Living, Playfulness and Sensory Processing in Children with Autism Spectrum Disorder: A Spanish Study. Children, 8(2), 61.
R1: We thank the Reviewer for his/her comments. We included in the Introduction the suggested references.
Q2: Some comments related to Participants section:
IRCCS ¿meaning?
R2: IRCCS is an acronym for “Istituto di Ricovero e Cura a Carattere Scientifico”, i.e., the institutional definition of Italian centers entitled to curry out both clinical and research activities. In the manuscript we now have spelled out such acronym.
Q3: Include more data on the location of the study. Country?
R3: All participants were Italian children recruited from different clinical centers.
Q4: Type of recruitment? Randomized, snowball, which?
R4: In line with our previous work on schoolers (Narzisi et al., 2022) our initial recruitment plan included two groups, namely ASD and TD. We computed the sample size for an ANCOVA with large effect size (0.4), alpha 0.05, power 0.9, 2 groups and 1 covariate. We obtained that 42 participants per group would have been suited for testing our hypothesis. We thus proceeded to first collecting the ASD children among those accessing the clinical centres and meeting the inclusion criteria reported in the manuscript, until we summed up 42 participants. Second, we collected data from the control sample recruiting kindergarten attendants matching the age of the ASD group. As the recruitment was made for entire kindergarten classes, we exceeded the envisioned number to avoid discarding children who already had accepted to participate to the study. Thus 56 TD children were included in the study. A preliminary analysis raised questions about the specificity of ASD-TD findings for the autistic disorders. To address this issue, we opted for an extra recruitment of children with a Diagnosis of Language Disorder (DLD) matched for age with the first two groups. The lower incidence of DLD in our partner clinical centres impeded us to reach the same numerosity as the first two groups, yet 18 children were enough to demonstrate different patterns between ASD and DLD. In the present version of the manuscript the non-homogenous sampling between ASD and DLD was included as limitation.
Q5: “Parents were given a brief questionnaire about their children's overall health condition and clinical history” It is not clear whether this phrase is for all parents or only for TD children.
R5: The Reviewer is right, this sentence could result misleading. It refers only to the parents of TD children. We modified this part, accordingly.
Q6: Related to DLD population: This population was not diagnosed based on the DSM-5? Clarify this point whether they did or did not. If they did not, avoid talking about DLD, and talk about language problems, without mentioning the diagnosis.
R6: The question is not fully clear to us. As reported in the manuscript the DLD participants had to be diagnosed according to DSM-5 (315.39 (F80.9)). We tried to make this point even clearer in the revised manuscript.
Q7: Was any analysis performed to confirm that the samples were age-matched?
R7: A one-way ANOVA was preliminarily conducted to confirm the homogeneity of our samples in terms of age. It did not reveal any significant difference among groups. This part was now included in the Results section.
Q8: First paragraph: All of these scales seem not to have been used in the study, so I suggest that they be eliminated. If the administration of any of them was used for some type of analysis that is not captured in the study, this analysis should be included.
R8: The Reviewer is right concerning the two cognitive tests, whose scores have not been used for the data analysis. Conversely, ADOS has at least to be mentioned when dealing with autistic children. Finally CARS and SCQ played a fundamental role not for the quantitative analysis, but to provide evidence of lack of ASD comorbidity for DLD children.
Q9: Data analysis section: Typographical error. Font size change in the middle of the paragraph. “and emotional problems, we applied linear regression analyses”
R9: Done
Q10: Figure 1: DLSD: error.
R10: Done
Q11: It would be more interesting to add a table with the numerical results of this analysis. They provide more data than graphs and help to better understand the differences.
R11: Done
Q12: Figure 2 Organize the wording of the text according to the organization of the images. Start the text with the first of the three graphics.
R12: Done
Q12: The limitations of the study have not been included
R12: The Reviewer is right, we have now added at the end of the Discussion a paragraph on the limitation of our study.
Q13: The conclusion should not introduce references. It should be the final point of the work, highlighting the main findings, establishing the connection with clinical practice. I suggest that it be rewritten. Part of the text that has citations could be included in the discussion. “one could develop individualized treatments according to each child's sensory profile, evaluating the adoption of multisensory strategies”. This statement denotes that the authors' knowledge of sensory integration therapy is shallow. Multisensory stimulation is a strategy that differs greatly from sensory integration, which is probably what the authors want to emphasize. I suggest they avoid talking about multisensory stimulation and talk about sensory integration therapy.
R13: Here the point is twofold. On one side the conclusion is structured exactly as the Reviewer suggests, recollecting the results and trying to establish a connection with clinical practice. Thus, respectfully, we do not feel it needs to be changed. The mere presence of references does not blur the nature of conclusion of the text, but is simply intended to quote support when statements refer to aspect not directly addressed by our study. That said, the Reviewer is right the imprecision of our former conclusion about the multisensory stimulation. As he/she indicated, we intended to refer to a general framework of sensory integration and not to multisensory stimulation. Following his/her suggestions, the revised manuscript refers only to sensory integration.
Q15: Review this section. There are some errors. View reference n24.
R15: Done

Reviewer 2 Report
Dear Authors:
Thank you for allowing me to read your interesting paper titled "Early sensory profile in autism spectrum disorders predicts emotional and behavioral issues" . In the title of the paper, they try to relate sensory processing to behavioural and emotional responses in autistic children. However, in the introduction they seem to focus on aberrant behaviours (citation 21-23) or simply maladaptive behaviours. There is evidence for a link between sensory processing dysfunction and the onset of anxiety in autistic children (e.g. Green et al 2010; Wigham et al. 2015) in particular the processing or identification of interoceptive sensory signals (e.g. Palser et al 2018). Perhaps worthy of comment and clarification in the bakcground section.
Otherwise, I think it is an excellent job. Congratulations.
Green SA, Ben-Sasson A. Anxiety disorders and sensory over-responsivity in children with autism spectrum disorders: is there a causal relationship? J Autism Dev Disord. 2010 Dec;40(12):1495-504. doi: 10.1007/s10803-010-1007-x.
Wigham S, Rodgers J, South M, McConachie H, Freeston M. The interplay between sensory processing abnormalities, intolerance of uncertainty, anxiety and restricted and repetitive behaviours in autism spectrum disorder. J Autism Dev Disord. 2015 Apr;45(4):943-52. doi: 10.1007/s10803-014-2248-x.
Palser,E.R. , Fotopoulou, A.Pellicano, E. and Kilner,J.M. (2018) The link between interoceptive processing and anxiety in children diagnosed with autism spectrum disorder: Extending adult findings into a developmental sample, Biological Psychology, Volume 136, Pages 13-21, https://doi.org/10.1016/j.biopsycho.2018.05.003.
Author Response
REVIEWER 2
Dear Authors:
Thank you for allowing me to read your interesting paper titled "Early sensory profile in autism spectrum disorders predicts emotional and behavioral issues" . In the title of the paper, they try to relate sensory processing to behavioural and emotional responses in autistic children.
Q1: However, in the introduction they seem to focus on aberrant behaviours (citation 21-23) or simply maladaptive behaviours. There is evidence for a link between sensory processing dysfunction and the onset of anxiety in autistic children (e.g. Green et al 2010; Wigham et al. 2015) in particular the processing or identification of interoceptive sensory signals (e.g. Palser et al 2018). Perhaps worthy of comment and clarification in the bakcground section.
Otherwise, I think it is an excellent job. Congratulations.
R1: We thank the Reviewer for his/her kindly comments. Following the suggestion we included in the new version of the manuscript the studies indicated enlarging the background section.
- Green SA, Ben-Sasson A. Anxiety disorders and sensory over-responsivity in children with autism spectrum disorders: is there a causal relationship? J Autism Dev Disord. 2010 Dec;40(12):1495-504. doi: 10.1007/s10803-010-1007-x.
- Wigham S, Rodgers J, South M, McConachie H, Freeston M. The interplay between sensory processing abnormalities, intolerance of uncertainty, anxiety and restricted and repetitive behaviours in autism spectrum disorder. J Autism Dev Disord. 2015 Apr;45(4):943-52. doi: 10.1007/s10803-014-2248-x.
- Palser, E.R. , Fotopoulou, A.Pellicano, E. and Kilner, J.M. (2018) The link between interoceptive processing and anxiety in children diagnosed with autism spectrum disorder: Extending adult findings into a developmental sample, Biological Psychology, Volume 136, Pages 13-21, https://doi.org/10.1016/j.biopsycho.2018.05.003.
